# Orthogonal regulation of phytochrome B abundance by stress-specific plastidial retrograde signaling metabolite

Jishan Jiang [1], Liping Zeng[1], Haiyan Ke [1], Brittenny De La Cruz[1] & Katayoon Dehesh [1]

Plant survival necessitates constant monitoring of fluctuating light and balancing growth demands with adaptive responses, tasks mediated via interconnected sensing and signaling networks. Photoreceptor phytochrome B (phyB) and plastidial retrograde signaling metabolite methylerythritol cyclodiphosphate (MEcPP) are evolutionarily conserved sensing and signaling components eliciting responses through unknown connection(s). Here, via a suppressor screen, we identify two phyB mutant alleles that revert the dwarf and high salicylic acid phenotypes of the high MEcPP containing mutant ceh1. Biochemical analyses show high phyB protein levels in MEcPP-accumulating plants resulting from reduced expression of phyB antagonists and decreased auxin levels. We show that auxin treatment negatively regulates phyB abundance. Additional studies identify CAMTA3, a MEcPP-activated calcium-dependent transcriptional regulator, as critical for maintaining phyB abundance. These studies provide insights into biological organization fundamentals whereby a signal from a single plastidial metabolite is transduced into an ensemble of regulatory networks controlling the abundance of phyB, positioning plastids at the information apex directing adaptive responses.

[1] Department of Botany and Plant Sciences, Institute of Integrative Genome Biology, University of California, Riverside, CA 92521, USA. Correspondence and requests for materials should be addressed to K.D. (email: kdehesh@ucr.edu)

Plant survival is dependent on light perception, required for photosynthetic activities and elicitation of biochemical and molecular modifications key to consequential adjustment of growth and development in a constantly changing environment. Phytochromes (phys) are a family of cytosolic photoreceptors (phyA to phyE) with varying extent of physiological functions, thus superimposing a layer of specialization on a foundation of partially overlapping functions[1–4]. Upon perception of red light, phys are activated and translocated to the nucleus, where they initiate and transduce photomorphogenesis signaling cascade. The main photoreceptor mediating photomorphogenesis in red light, however, is phyB. Red light activated phyB is imported into the nucleus and forms phyB-containing nuclear bodies (phyB-NBs)[3,5]. Formation of phyB-NBs depends on binding to and sequestration of the basic helix-loop-helix (bHLH) transcription factors, Phytochrome Interacting Factor 1 (PIF1), PIF3, PIF4, PIF5, and PIF7[6,7]. This binding facilitates phosphorylation and degradation of PIFs by the 26S proteasome and reduces their capacity to bind their DNA targets to initiate responses[6,8,9].

Shade avoidance response (SAR) is a central physiological and developmental response to the light signals resulting from changes in the ratio of red (R):far-red (FR) light caused by neighboring vegetation leading to among others enhanced stem and petiole growth as a mechanism attempting to reach an strata with more available light[10–12]. Although several phys contribute to SAR, phyB plays a particularly prominent role, as evidenced by the display of classical SAR phenotypes in phyB mutants[10,11]. The SAR-mediated enhanced growth can also be simulated by exogenous application of auxin as well as through genetic manipulations such as generation of lines with increased auxin levels[13–17]. It is well established that PIF4, PIF5, and PIF7 play a major role, whereas PIF1 and PIF3 assume a minor role in mediating SAR, and additional evidence indeed illustrate that PIFs target promoter elements of multiple auxin biosynthetic and signal transduction genes[8,18–20].

The overall function of phys in transcriptional regulation conveys exquisite light control of photosynthetic machinery in chloroplast, followed by consequential physiological responses. However, the success of these biological processes depends on tight coordination and dynamic alignment of cellular responses to light perception through activation of signal transduction pathways essential for fine-tuning of inter-organellar communication, specifically through chloroplast-to-nucleus signaling, a process called retrograde signaling[21].

Our search for identification of a stress specific retrograde signal, led to the discovery of the plastid-derived metabolite 2-C-methyl-d-erythritol 2,4-cyclodiphosphate (MEcPP), that functions both as a precursor of isoprenoids produced by the conserved and essential plastidial methylerythritol phosphate (MEP) pathway, and as a stress-specific retrograde signaling metabolite that communicates plastidial perturbations to the nucleus[22]. This discovery was founded on a genetic screen that led to the isolation of a mutant line designated ceh1, for constitutive expression of Hydroperoxide Lyase (HPL), an otherwise stress-inducible nuclear gene encoding a plastidial enzyme in the HPL branch of the oxylipin pathway[22]. This mutation results in accumulation of high MEcPP levels and the consequent growth retardation in concert with induction of selected stress-responsive nuclear genes and their respective metabolites. Among the nuclear genes with enhanced expression level in ceh1 is isochorismate synthase 1 (ICS1), a stress-inducible nuclear gene encoding a key plastidial enzyme in the salicylic acid (SA) biosynthetic pathway[22,23]. Furthermore, we established that MEcPP transduces signals to activate a core set of general stress response genes through induction of a transcriptionally centered stress hub, activated predominantly by the function of a member of the calmodulin-

binding transcriptional activator family, CAMTA3[24]. In addition, we demonstrated that MEcPP controls adaptive growth by modulating auxin levels and distribution patterns in response to stress[25].

In this study, to identify the genetic components of MEcPP signal transduction pathway we report the results of a suppressor screen for revertants with fully or partially abolished ceh1 phenotypes in spite of high MEcPP levels. This approach led to the identification of two independent phyB mutant alleles that notably reverted high SA levels and dwarf stature phenotypes of the ceh1 mutant. Subsequent analyses of these mutant lines identified the network of MEcPP-mediated signaling cascade that alter phyB protein level and uncovered the role of auxin and CAMTA3 in maintaining phyB abundance. Moreover, pharmacological approaches illustrated that the likely function of CAMTA3 is to stabilize phyB protein level through impairment of proteasome-mediated degradation machinery. We conclude that phyB is a component of MEcPP-signal transduction pathway and that MEcPP-mediated regulation of phyB level is multifaceted. By extension this sheds light on the links between the two evolutionarily conserved sensing and signaling cascades essential for adjustment of growth and development in a constantly changing environment.

## Results

**PhyB is a component of MEcPP signal transduction pathway.** To identify the genetic components of the plastidial retrograde signal transduction pathway, we mutagenized the high MEcPP-accumulating ceh1 in search of revertants that despite high MEcPP levels display full or partial recovery of the mutant dwarf stature with either no and/or reduced expression of HPL and/or the ICS1-derived product SA. This led to identification of two independent revertants (41-16 and 56-6) with classical SAR phenotypes, namely seedlings with long hypocotyls and long petioles[10–12], and overall a larger statue than that of the ceh1 mutant, albeit still smaller than the parent wild-type (P) seedlings (Fig. 1a, b and Supplementary Fig. 1a). Furthermore, measurements of MEcPP show a ~50% reduction of the metabolite levels in the revertants as compared with the ceh1, yet with ~200-fold higher levels than that of the P seedlings (Fig. 1c). Following analyses showed notable (~70%) reduction in SA levels of the revertants as compared with ceh1 (Fig. 1d). Subsequent sequencing analyses identified two different single non-sense mutations in the PHYB resulting in premature terminations of the protein in revertants (Fig. 1e). Complementation studies using the wild type PHYB coding region under the control of the native promoter reestablished the ceh1 mutant phenotypes in the revertants, and conversely the double mutant (ceh1/phyB-9) generated from crosses between ceh1 and a phyB null mutant (phyB-9) mimicked the growth phenotypes and the SA levels of the revertants (Fig. 1a–d). The data verifies revertants as phyB mutant alleles.

To examine phyB general and/or specific function in MEcPP-mediated signal transduction pathway, we also compared expression of HPL, a stress-inducible gene constitutively expressed in the ceh1, in aforementioned genotypes. Comparable HPL expression levels in ceh1 and ceh1/phyB-9 seedlings excluded the involvement of phyB in the MEcPP-mediated induction of HPL (Supplementary Fig. 1b).

The data collectively establishes phyB as a component of MEcPP signal transduction pathway involved in regulation of growth and SA level, but not in the induction of HPL expression.

**PhyB protein abundance in the MEcPP-accumulating lines.** To address a potential transcriptional/posttranscriptional alteration of phyB in MEcPP accumulating plants, we initially examined the

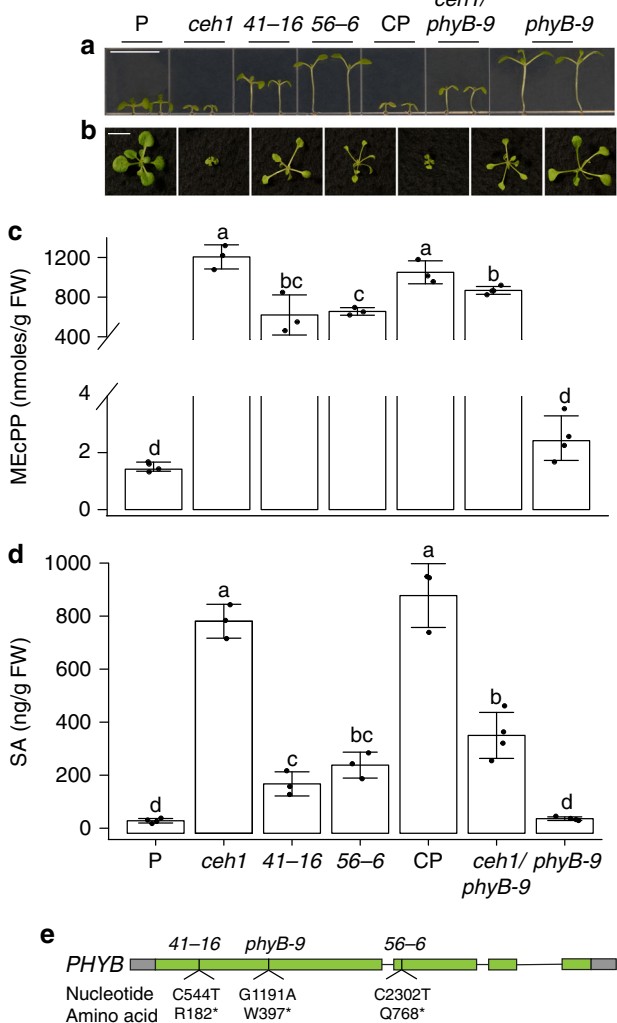

**Fig. 1** PhyB is a component of MEcPP signal transduction pathway. **a**, **b** Representative images of 1-week-old and 2-week-old P, *ceh1*, *41-16*, *56-6*, complementation line (CP), *ceh1/phyB-9*, and *phyB-9* seedlings grown in long-day (LD; 16 h light/8 h dark), respectively. **c** MEcPP measurements of 2-week-old seedlings of aforementioned genotypes. **d** Measurements of Salicylic Acid (SA) levels of samples used in (**c**). **e** Schematic illustration of *PHYB* gene. Green boxes represent exons, gray boxes represent UTRs, the actual nucleotide and amino acid changes of the mutants are displayed underneath. Data are mean ± SD, each genotype with at least three biological replicates. Letters above bars indicate significant differences determined by Tukey's HSD method ($P < 0.05$). Scale bar: 1 cm

*PHYB* transcript levels in *ceh1* seedlings. Similar *PHYB* transcript levels in *ceh1* and P plants excludes a MEcPP-mediated transcriptional regulation of the photoreceptor (Fig. 2a). Next, we examined phyB protein levels by immunoblot analyses of total proteins extracted from dark and long-day (LD; 16 h light/8 h dark) grown P and *ceh1* seedlings. These results showed similarly high phyB protein levels in the dark grown P and *ceh1* genotypes (Fig. 2b). Intriguingly, we observed the expected reduction in phyB protein of P seedlings grown under LD whereas, under the same growth conditions the *ceh1* seedlings maintained comparable levels of phyB protein to that present in the dark grown seedlings (Fig. 2b).

Total protein extract represents biologically active (nuclear) and biologically inactive (cytosolic) forms of phyB. To

differentiate between the two states of phyB, we examined phyB protein abundance of nuclear protein extracts by immunoblot analyses. The outcome mimicked the total protein extract data, and further confirmed the presence of higher phyB protein levels in the nuclear extract of LD grown *ceh1* as compared with P (Fig. 2c).

To establish a potential link between MEcPP and phyB protein abundance, we performed immunoblot analyses on protein extracts of DEX-inducible MEcPP accumulating line (*HDSi*) and the control P plants, before and at different hours post induction. The increased phyB levels in the induced *HDSi* seedlings establishes a correlation between accumulation of MEcPP and enhanced phyB protein abundance (Fig. 2d and Supplementary Fig. 1c).

In addition to MEcPP, *ceh1* mutant accumulates substantial amount of the defense hormone, salicylic acid (SA)[22,26]. The reported phyB role in SA accumulation and signaling[27] prompted us to examine the potential reciprocal role of this defense hormone in regulation phyB protein abundance, by employing the previously generated SA deficient double mutant line, *ceh1/eds16*[22]. Immunoblot analyses on proteins extracts of P, *ceh1*, *eds16-1*, and *ceh1/eds16-1* clearly demonstrate comparable phyB protein levels in *ceh1* and *ceh1/eds16* (Supplementary Fig. 2a). This confirms that phyB abundance in the *ceh1* mutant is SA-independent.

Next, we tested biological ramification of over-accumulation of phyB in *ceh1* by fluence response studies carried out by measurements of hypocotyl lengths of seedlings grown in dark and continuous red-light (Rc) with increasing fluence rate (light intensity) (Fig. 3a). Dark grown P and *ceh1* seedlings exhibit similar hypocotyl lengths, whereas under Rc these genotypes display differentially reduced hypocotyl growth. The progressively shorter hypocotyl length of *ceh1* with increasing fluence rate of red-light as compared with the P seedlings suggests photosensory hypersensitivity of *ceh1* to Rc, most likely due to enhanced phyB protein abundance (Fig. 3a). This notion was further tested by examining hypocotyl length of seedlings grown in continuous white light (CL), and short-day (SD; 8 h light/16 h dark) conditions. The statistically significant lower relative SD/CL hypocotyl length in *ceh1* as compared with P implies the presence of a more active (Pfr) form of phyB in the *ceh1* relative to the relaxed (Pr) form (Fig. 3b).

This data collectively establishes the tight correlation between MEcPP accumulation and enhanced phyB protein abundance, ultimately resulting in photo-sensory hypersensitivity of *ceh1* to red light.

***PIF4* and *PIF5* overexpression mimic revertants phenotypes.**
PIFs are known antagonist of phys, and established agonists of SAR response[8,10]. Amongst the seven PIF family members, PIF4 and PIF5 have been shown unequivocally to mediate this response[8,10,14]. Moreover, phyB-PIF interaction reciprocally induces phyB degradation, in a mutually-negative, feedback-loop configuration[9,28]. To examine the potential role of PIFs in the *ceh1* transduction pathway, we examined expression levels of *PIF1*, *3*, *4*, and *5*, of which, *PIF4* and *PIF5* expressions were reduced in *ceh1* as compared with P (Fig. 4a and Supplementary Fig. 2b). The slight reduction (25–30%) in the *PIF4* and *5* expression levels led us to genetically assess their involvement in the MEcPP-mediated signaling cascade by generating *ceh1* over-expressing *PIF4* and *PIF5* lines. Subsequent analyses display partial recovery of *ceh1* dwarf phenotype in the overexpresser backgrounds, with more pronounced growth in *ceh1/PIF5-OX* compared with *ceh1/PIF4-OX* seedlings (Fig. 4b). This differential growth correlates well with the respective MEcPP content,

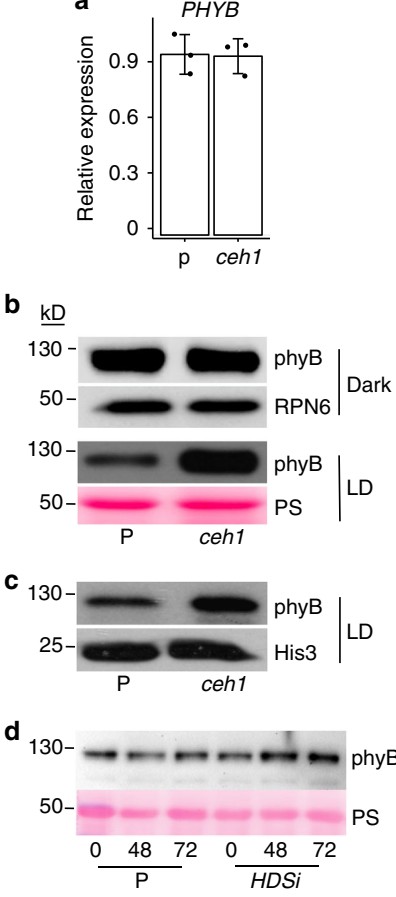

**a** PHYB

**b** kD

130 — phyB
50 — RPN6 ⎤ Dark

130 — phyB
50 — PS ⎤ LD

P          ceh1

**c**
130 — phyB
25 — His3 ⎤ LD

P          ceh1

**d**
130 — phyB
50 — PS

0   48   72   0   48   72
—— P ——   —— HDSi ——

**Fig. 2** PhyB protein abundance in MEcPP-accumulating lines. **a** PHYB expression level is not altered in ceh1 as compared with P. Total RNAs were isolated from 7-day-old seedlings of P and ceh1 grown under long-day (LD; 16 h light/8 h dark) condition and subjected to qRT-PCR analysis. The mRNA levels of PHYB gene was normalized to the levels of At4g26410 (M3E9). Data are mean ± SD of three biological replicates and three technical replicates. Two-tailed Student's t tests confirms no significant differences of PHYB expressions between P and ceh1. **b** PhyB protein abundance in dark and LD grown P and ceh1 seedlings. Total proteins extracted from 7-day-old dark and LD grown P and ceh1 seedlings were subjected to immunoblot analyses using phyB antibody. Immunoblot using RPN6 antibody in concert with Ponceau S (PS) staining were used as loading controls for dark and LD grown seedlings, respectively. **c** Immunoblot analyses of phyB protein levels in nuclear protein extracts from LD grown 7-day-old P and ceh1 seedlings. Blots probed with Histone 3 (His 3) antibody show equal loading of nuclear proteins. **d** PhyB protein levels in 7-day-old P and DEX-inducible MEcPP-accumulating HDSi seedlings before (0) and hours (48 and 72) post induction. Ponceau S (PS) staining was used as the protein loading control

namely the presence of ~20% less MEcPP in ceh1/PIF5-OX than in ceh1 or ceh1/PIF4-OX lines (Fig. 4c). Next, we analyzed SA levels and determined a ~60–65% reduction in the SA content of ceh1/PIF4 and 5 overexpresser lines compared with the ceh1 mutant background (Fig. 4d).

The recovery of ceh1 dwarf phenotype in PIF4 and PIF5 overexpressing seedlings led us to examine the phyB protein levels in the respective lines by immunoblot analyses. The data show the expected reduced phyB abundance in ceh1/PIF4-OX and ceh1/PIF5-OX as compared with ceh1, and the expected lower abundance in both OX seedlings relative to P plants. It is of note that phyB relative abundance in ceh1/PIF4-OX is higher than that

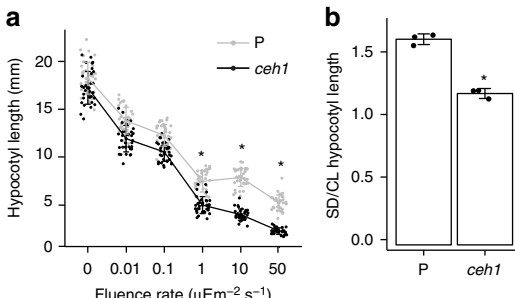

**a** **b**

**Fig. 3** Photosensory hypersensitivity of ceh1. **a** Seeds of P and ceh1 were germinated and grown under continuous red light of various fluence rate (0, 0.01, 0.1, 1, 10, and 50 μEm$^{-2}$ s$^{-1}$) for 7 days before hypocotyl length measurements. Dark grown (fluence rate 0) seedlings were used as control. **b** P and ceh1 seeds germinated and grown under short-day (SD; 8 h light/16 h dark) or continuous white light (CL) for 7 days were subsequently subjected to hypocotyl length measurements. Data in (**a**) and (**b**) are mean ± SD, each with three biological replicates and 45 seedlings per replicate. Asterisks denote significant differences as determined by a two-tailed Student's t tests with a significance of $P < 0.05$

of ceh1/PIF5-OX, a phenotype consistent with their differential growth phenotypes, namely larger ceh1/ PIF5-OX seedlings (Fig. 4b, e and Supplementary Fig. 2c).

This data alludes to high phyB level in ceh1, in part due to reduced PIF4 and 5 transcript levels, and further illustrates a dominant role of PIF5 in MEcPP-mediated regulation of phyB abundance.

**Auxin treatment reduces phyB protein abundance in ceh1.** The established role of PIF4 and 5 in transcriptional regulation of auxin biosynthetic and signal transduction genes through targeting their promotors[8,18,19,29], together with our report on MEcPP-mediated reduction of auxin level and auxin transport[25], prompted us to examine a potential involvement of auxin in regulation of phyB protein abundance in ceh1. Thus, we monitored growth phenotypes, and phyB protein levels of P and ceh1 seedlings grown in the absence (mock) and the presence of exogenously applied auxin (Fig. 5a, b). The result corroborate our previous finding[25], and further establish that auxin induction of hypocotyl growth is in concert with reduced phyB levels in all auxin treated seedlings, most notably in the ceh1 mutant background. This data was further confirmed by genetic approaches using double mutant lines of ceh1 introgressed into auxin receptor mutant tir1-1 (ceh1/tir1-1) known to be deficient in a variety of auxin-regulated growth processes including hypocotyl elongation and normal response to auxin[30]. Subsequently, we examined growth phenotypes and phyB protein abundance in P, ceh1, ceh1/tir1-1, and tir1-1 seedlings grown in the presence and absence of auxin (Fig. 6a, b). This data clearly shows that auxin treatment is notably less effective in promoting hypocotyl growth in ceh1/tir1-1 as compared with the ceh1 mutant. Notably, auxin effect on reducing phyB abundance is attenuated in tir1-1 backgrounds as compared with the respective parent genotypes.

Collectively, these results strongly support the negative feedback regulatory function of auxin in regulating phyB protein abundance.

**PhyB protein abundance in ceh1 is CAMTA3-dependent.** We have previously established that MEcPP-mediates induction of general stress response genes via activation of calmodulin-binding transcriptional activator 3 (CAMTA3), in a calcium dependent manner[24]. In addition, we previously showed and here

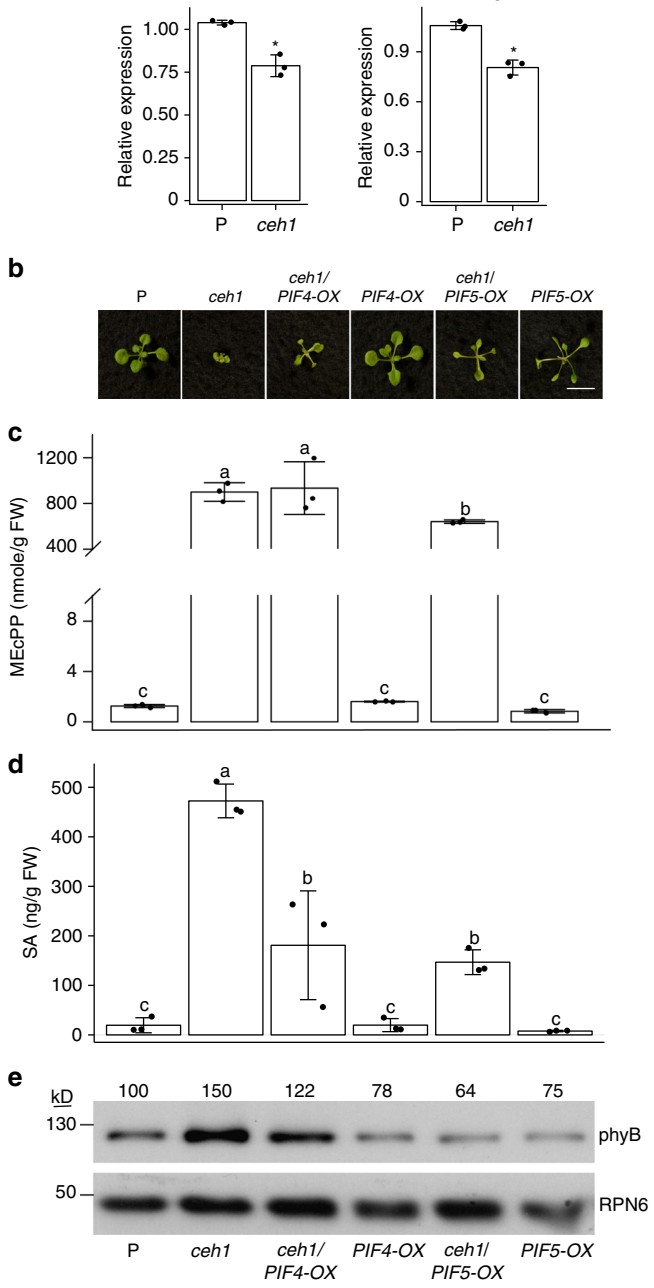

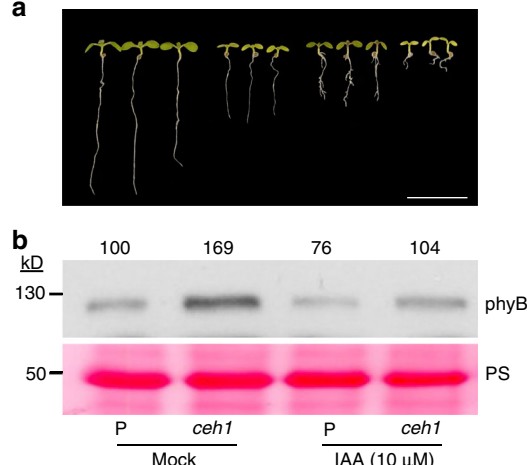

**Fig. 5** Auxin treatment reduces phyB protein in *ceh1*. **a** Representative images of 7-day-old LD (16 h light/8 h dark) grown P and *ceh1* seedlings in the absence (mock, 0.01% ethanol) and the presence of IAA. Scale bar: 1 cm. **b** Abundance of phyB protein in total protein extracts of P and *ceh1* seedlings grown in the absence (mock) and presence of IAA. Numbers above the immunoblot image represent normalized phyB abundance in mock and IAA treated seedlings relative to P plants. Ponceau S (PS) staining was used as the loading control

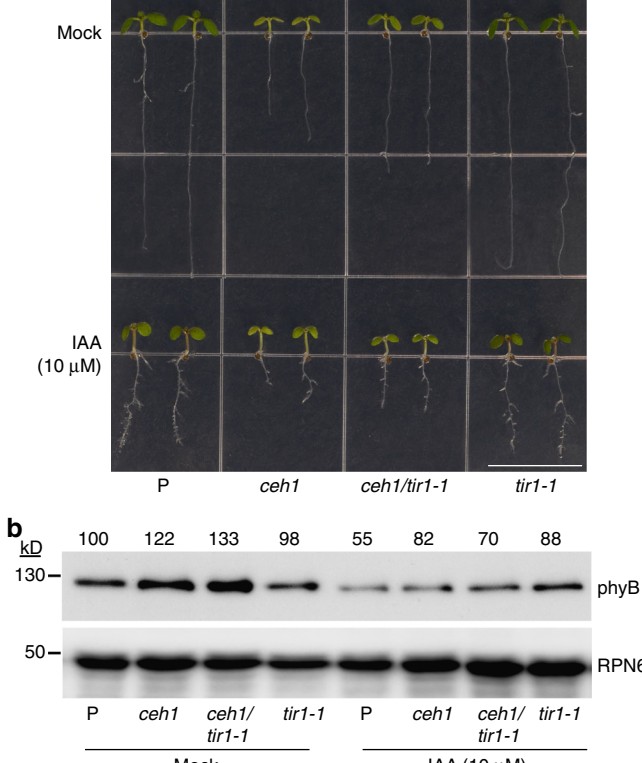

**Fig. 6** Auxin regulation of phyB abundance is attenuated in *tir1-1*. **a** Representative images of 7-day-old LD (16 h light/8 h dark)grown P, *ceh1*, *ceh1/tir1-1*, and *tir1-1* seedlings in the absence (mock, 0.01% ethanol) and the presence of IAA. Scale bar: 1 cm. **b** Abundance of phyB protein in total protein extracts of P, *ceh1*, *ceh1/tir1-1*, and *tir1-1* seedlings in the absence (mock) and presence of IAA. RPN6 was used as the loading control. Numbers above the immunoblot image represent normalized phyB abundance in the examined genotypes relative to P seedlings

**Fig. 4** *PIF4* and *PIF5* overexpression mimics *phyB* revertants phenotypes. **a** Expression levels of *PIF4* and *PIF5* in P and *ceh1* seedlings. Experiments were performed as described in Fig. 2a. Asterisks denote significant differences as determined by a two-tailed Student's *t* tests with a significance of *P* < 0.05. **b** Representative images of 2-week-old P, *ceh1*, *ceh1/PIF4-OX*, *PIF4-OX*, *ceh1/PIF5-OX*, and *PIF5-OX* seedlings. Scale bar: 1 cm. Levels of MEcPP, with the white line indicating a change of scale on the *Y* axis (**c**) and SA (**d**) in the aforementioned genotypes. Data are mean ± SD of each genotype with three replicates. Letters above bars indicate significant differences determined by Tukey's HSD method (*P* < 0.05). **e** Abundance of phyB in total protein extracts of P, *ceh1*, *ceh1/PIF4-OX*, *PIF4-OX*, *ceh1/PIF5-OX*, and *PIF5-OX* seedlings. Immunoblots were carried out as described in Fig. 2b, with RPN6 as the protein loading control. Numbers above the immunoblot image represent normalized phyB abundance in the genotypes examined relative to P plants

reconfirmed the partial size recovery of *ceh1* dwarf stature in the *ceh1/camta3* double mutant lines[24] (Fig. 7a). This result prompted us to perform RNA seq analyses on *camta3* mutant lines in P and *ceh1* mutant backgrounds. Bioinformatics analyses of the data determined that the expression of ~90% of repressed genes in *ceh1* are CAMTA3-dependent, and are significantly overrepresented by genes in light- and auxin-response pathways (Supplementary Fig. 3a and Supplementary Data 1).

This data prompted us to extend the bioinformatics analyses and compare these CAMTA3-dependent genes with those IAA[31] or PIF[32] regulated genes (Supplementary Fig. 3b, c and Supplementary Data 2 and 3). Consistent with the GO term analysis (Supplementary Fig. 3a), among the CAMTA3-suppressed genes, ~9.6% and ~10.4% genes are overrepresented in light- and auxin-response pathways induced by PIF and IAA, respectively. We also compared the microarray dataset of constitutively active allele of phyB (*YHB*)[33], with CAMTA3-dependent genes and found that ~17.3% CAMTA3-induced genes are also upregulated in *YHB*, and ~8.1% CAMTA3-suppressed genes are also downregulated in *YHB* (Supplementary Fig. 3d, e and Supplementary Data 4 and 5). These data illustrate an overlap between gene regulatory network regulated by MEcPP and those controlled by CAMTA3, PIF, phyB, and auxin.

This data in concert with our previous observations[24] followed by reconfirmation of partial recovery of *ceh1* dwarf phenotype in a *ceh1/camta3* double mutant (Fig. 7a) prompted us to examine the likelihood of CAMTA3 involvement in regulation of phyB in the *ceh1* background. Initial expression analyses show insignificant differences in the *PHYB* transcript levels in examined genotypes (P, *ceh1*, *ceh1/camta3*, and *camta3*) (Supplementary Fig. 4a). Next, we examined phyB protein abundance in these genotypes (Fig. 7b and Supplementary Fig. 4b). Clear reduction of phyB protein abundance in *ceh1/camta3* to the basal levels present in P and *camta3* supports the notion of the regulatory function of CAMTA3 in maintaining high phyB protein levels in the *ceh1* background.

Given the established dependency of CAMTA3 functionality on $Ca^{2+}$ together with the proposed function of MEcPP as a rheostat for release of $Ca^{2+}$ for activation of CAMTA3[24], we examined phyB protein abundance in P and *ceh1* in the presence and absence (mock) of lanthanum chloride (LaCl₃; $Ca^{2+}$ channel blocker)[34]. The data illustrate reduced phyB protein levels in seedlings treated with this agent most notably in the *ceh1* mutant as compared with the mock treated seedlings (Fig. 7c, d and Supplementary Fig. 4c). The result collectively supports regulatory function of CAMTA3 in stabilizing phyB protein levels in a $Ca^{2+}$-dependent manner.

**Bortezomib stabilizes phyB protein level in *ceh1/camta3*.** To explore mode of CAMTA3 action in stabilizing phyB protein levels, we examined phyB protein levels in LD grown P, *ceh1*, *ceh1/camta3*, and *camta3* seedlings treated with a 26S proteasome inhibitor, bortezomib[35]. The representative blots clearly show higher phyB protein levels in all bortezomib treated seedlings versus the respective mock treated seedlings (Fig. 8a–c and Supplementary Fig. 5a–c). It should be noted that both presented blots consistently display higher phyB protein levels in the presence of proteasome inhibitor albeit at different degrees, most likely due to varying bortezomib penetration rate. This increase is most notable in *ceh1/camta3* and *camta3* bortezomib treated seedlings versus untreated plants. The apparent variations in phyB protein abundance between untreated and mock treated seedlings (Fig. 8a–c and Supplementary Fig. 5a–c) is most likely due to growth conditions requiring transfer of seedlings into liquid media for mock treatment versus

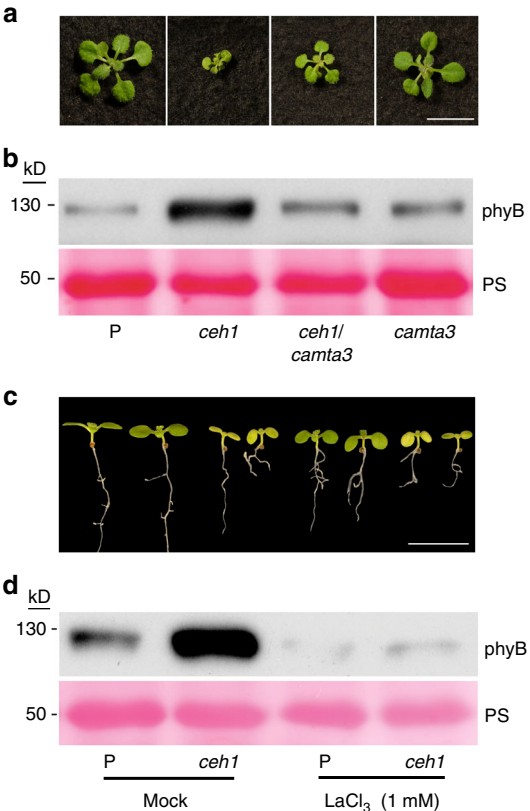

**Fig. 7** PhyB protein abundance in *ceh1* is CAMTA3- and $Ca^{2+}$-dependent. Representative images (**a**) and phyB protein levels (**b**) in 7-day-old LD (16 h light/8 h dark) grown P, *ceh1*, *ceh1/camta3*, and *camta3* seedlings. Representative images (**c**) and phyB protein levels (**d**) in 7-day-old P and *ceh1* seedlings in the absence (mock, 0.01% ethanol) and presence of LaCl₃, respectively. Ponceau S (PS) staining was used as loading control. Scale bar: 1 cm

maintaining the untreated seedlings on plates, as described in the Method section.

Collectively, the result strongly supports the notion of CAMTA3 function in stabilizing phyB protein level by impairing proteasome-mediated degradation machinery.

## Discussion

The photoreceptor phyB and the plastidial retrograde signaling metabolite MEcPP are two evolutionarily conserved sensing and signaling components essential for integrity of plants in ever changing environment. The data presented here reveal a functional link between these two essential sensing and signaling pathways and uncovers phyB as component of MEcPP signal transduction pathway, and further identifies MEcPP-mediated multifaceted regulatory network responsible for increased phyB protein abundance and the respective physiological ramifications such as aligning growth and development with stress responses in a constantly changing light environment.

Here, we specifically identify two independent suppressor lines, each with a non-sense mutation in *PHYB*, displaying a partial reversion of the dwarf phenotype of the *ceh1* mutant. In addition, we demonstrate comparable phyB protein in light and dark grown seedlings of constitutively and inducible MEcPP-accumulating lines, as opposed to light-induced reduction of phyB levels in the control line. This is an unexpected finding since

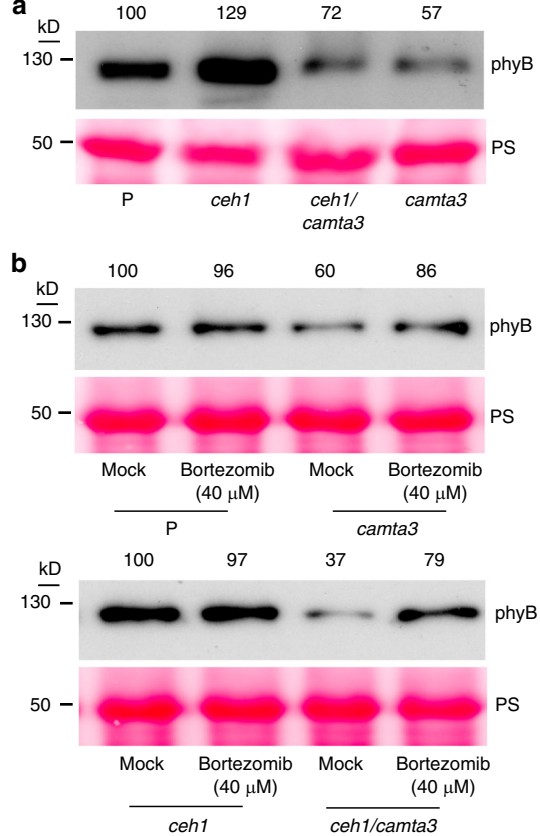

**Fig. 8** CAMTA3 stabilizes phyB protein abundance. **a** phyB protein abundance in nontreated 7-day-old LD grown P, *ceh1*, *ceh1/camta3*, and *camta3* seedlings. **b** phyB protein abundance in 7-day-old LD (16 h light/8 h dark) grown P, *ceh1*, *ceh1/camta3*, and *camta3* seedlings in the absence (mock, 0.1% DMSO) and presence of the 26S proteasome inhibitor, bortezomib. Ponceau S (PS) staining was used as loading control. Numbers on top of the immunoblot images represent normalized phyB abundance in various genotypes and in response to bortezomib treatment relative to the P (the top two panels) or *ceh1* (the bottom panel) seedlings

light mediates re-localization of cytosolic phyB to nucleus, and subsequent induction of a mutually regulatory intermolecular transaction between phyB and PIF transcription factor leading to rapid degradation of both proteins, and concomitant negative feedback modulation of phyB levels in Arabidopsis[36]. High abundance of phyB in light grown *ceh1* seedlings could contribute to stunted phenotype of the mutant as evidenced from earlier reports on the growth retardation of Arabidopsis plants over-expressing phyB[37]. The role of phyB in regulating growth and development is reported to be through repressing auxin response genes as well as through regulation of both SUR2, a suppressor, and TAA1, an enhancer of IAA biosynthesis[38,39]. Intriguingly, SUR2 (RED1/ATR4) is a cytochrome P450 monooxygenase that moderates the balance between IAA and plant defense compounds, indole glucosinolates[40,41]. This data together with high and selective production of indole glucosinolates in the *ceh1* mutant leads to the assumption that MEcPP-mediated induction of this defense metabolite is likely via enhanced abundance of phyB leading to modulation of auxin levels and by extension indole glucosinolates in plants[42]. In fact regulatory function of phyB in plant defense responses is also evident from enhancement of anti-insect resistance via functional integration of phyB with jasmonates, likely through a conserved hardwire transcriptional network that attenuates growth upon activation of

jasmonate signaling[43]. Importantly, reduced SA levels in suppressor lines as well as in *ceh1/phyB-9* double mutant compared with the levels in *ceh1* background is a clear testimony to the role of phyB in regulating defense hormones.

The next question is the nature of MEcPP-targeted regulatory processes that stabilize phyB levels in light. The schematic model (Fig. 9) depicts MEcPP-mediated regulation of phyB abundance via multiple intersecting pathways, amongst them reduced expression of *PIF4* and *5* contributing to phyB stability, in conjunction with reduced auxin known to be regulated by PIFs[8,18,29]. Auxin is also controlled by MEcPP-mediated reduction of the levels and distribution of this hormone by dual transcriptional and posttranslational regulatory inputs[25]. In addition to the known phyB regulation of auxin, here we establish the reciprocal and negative feedback regulatory function of auxin in adjusting phyB protein abundance as evidenced by reduced phyB levels in seedlings treated exogenously with auxin, and notable attenuation of the response in genotypes in the auxin receptor mutant (*tir1-1*) background. This data together with earlier reports of auxin regulation by phyB establish a mutually negative, feedback-loop between phyB and auxin.

Importantly, we show CAMTA3-dependent maintenance of high phyB protein abundance. Previously established comparable auxin levels in *ceh1* and *ceh1/camta3* discard the potential action of CAMTA3 through alteration of auxin levels[24]. As such, we propose that function of CAMTA3 is likely via impairment of proteasome-mediated degradation of the light-activated photoreceptor proteasome COP1-mediated proteolysis of phyB. Indeed, the established phyB degradation mediated by COP1 E3 ligase[44], and impairment of phyA degradation in the presence of sucrose[45], offer a potential link between carbon source and COP1-mediated proteolysis of phyB in *ceh1*.

Pharmacological application of a proteasome inhibitor bortezomib resulting in stabilization of phyB protein levels, most notably in *camta3* mutant backgrounds, strongly supports a direct or indirect function of CAMTA3 in stabilizing phyB protein levels by impairing proteasome-mediated degradation machinery.

In summary, this report unlocks the complexities of plastidial regulated growth-defense tradeoffs and specifically identifies the stress-specific plastidial retrograde signaling metabolite, MEcPP, as an orthogonal regulatory hub of photomorphogenesis through regulation of intersecting pathways of CAMTA3-based transcriptional network and auxin signaling cascade, which collectively control phyB abundance. Delineation of this exquisitely tightly controlled circuitry via plastidial retrograde signaling provides a prime example of biological organization whereby a specific plastidial signal such as MEcPP is transduced into a repertoire of biological responses that collectively regulate photomorphogenesis, a central adaptive strategy critical for securing organismal integrity. Furthermore, this work provides insight into governing rules of complex biological systems and identifies plastids not only as the metabolic hub but also as a center for integration of sensory and signaling networks central to plant adaptive responses to the prevailing environment.

## Methods

**Plant materials.** Arabidopsis plants used in this study were all in Columbia-0 background. The control, parent line (P) was transgenic line expressing *HPL::LUC* construct initially employed for EMS mutagenesis and isolation of *ceh1* mutant. *tir1-1* (CS3798) was ordered from ABRC, and *ceh1/tir1-1* double mutant was generated by crossing the two mutant lines. *PIF4-OX* and *PIF5-OX* were gifts from Dr. Peter Quail (University of California, Berkeley).

**Suppressor screen.** EMS mutagenesis was performed on seeds (~200 mg) of *ceh1* mutant lines. Seeds were treated overnight with 0.2% (v/v) ethylmethylsulfonate

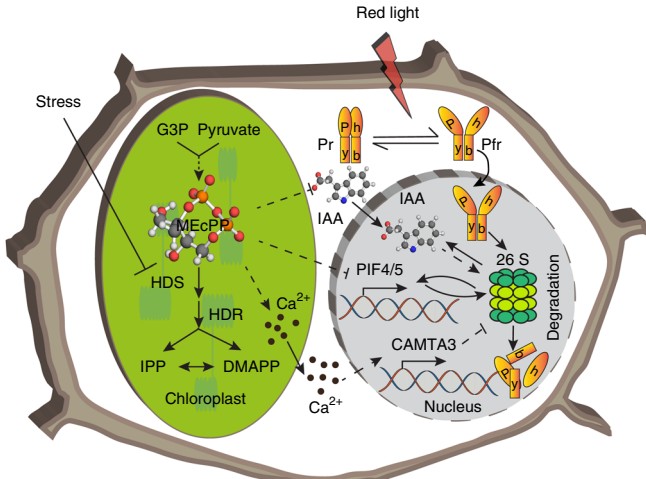

**Fig. 9** Orthogonal regulation of phyB protein abundance by MEcPP. Stress interference with HDS enzyme activity and the consequential accumulation of MEcPP transduces intersecting signaling circuitries that regulate abundance of the red-light photoreceptor, phyB, via auxin signaling cascade and $Ca^{2+}$-dependent CAMTA3-based transcriptional network. The likely function of CAMTA3 is to stabilize phyB protein levels through impairment of proteasome-mediated degradation of the light-activated phyB

(EMS), followed by three washes with 0.1 M sodium thiosulfate and subsequent eight rinses with water. The M1 plants were grown on soil under long-day (LD; 16 h light/8 h dark) condition at 22 °C. The M2 seeds were harvested and bulked into 206 pools by pots. Around 100 M2 plants from each pool were screened for mutants with fully or partially bigger size than that of *ceh1*. The selected M2 plants were grown for two additional generations and only the mutants with inheritable reverted *ceh1* growth phenotype were selected, amongst them *41-16* and *56-6* lines. The long hypocotyl and long petiole of these two revertants were reminiscent of the established phyB mutant phenotypes. This led us to directly sequence *PHYB* and identify nonsense mutations responsible for early termination of coding regions of the two alleles. Complementation studies confirmed phyB as the suppressor of *ceh1*.

**LaCl₃ and IAA treatments**. Surface-sterilized seeds were planted on half strength Murashige and Skoog medium (1/2 MS: 2.2 g/L Murashige and Skoog salts, pH 5.7, and 8 g/L agar), stratified at 4 °C for 3 days, and germinated under LD for 4 days prior to their transfer onto plates containing 1/2 MS medium together with mock control (0.01% ethanol), $LaCl_3$ (1 mM), and IAA (10 μM) for 3 additional days prior to analyses.

**Bortezomib treatment**. Surface-sterilized seeds were planted on 1/2 MS plates, stratified at 4 °C for 3 days, grown under LD for 7 days. These seedlings were then transferred to liquid 1/2 MS media containing mock control (0.1% DMSO) or bortezomib (40 μM) dissolved in 0.1% DMSO for 8 h on a shaker in the light chamber[35].

**Light curve of continuous red light**. Surface-sterilized seeds were planted on 1/2 MS plates, stratified at 4 °C for 5 days, and subsequently germinated and grown under continuous red light at various fluence rate for 7 days prior to hypocotyl measurement. Hypocotyl measurement was performed using image J[25].

**Quantitative RT-PCR**. The RNA extraction and data normalization were performed as previously described[25]. Gene-specific primers were designed using QuantPrime q-PCR primer design tool (http://www.quantprime.de/) and listed in Supplementary Table 1. Each experiment was performed with at least three biological and each with three technical replicates.

**Protein extraction and immunoblot analyses**. Total and nuclear fraction protein extractions were performed according to reported protocol with minor modifications[46]. Specifically, 0.2 g of 7-day-old seedlings were collected, ground with liquid nitrogen, homogenized in extraction buffer (10 mM Hepes, pH 7.6, 1 M Sucrose, 5 mM KCl, 5 mM $MgCl_2$, 5 mM EDTA, 14 mM 2-ME, 0.4% Triton X-100, 0.4 mM PMSF, 20 μM MG132, 20 μM MG115, and Proteinase Inhibitor). Crude extract was filtered through Miracloth and saved as total protein. For nuclear fractionation, after filtration, the crude extract was set on ice for 10 min, loaded on top of 15% percoll and centrifuged at 3000 × *g* for 10 min at 4 °C. Upon removal of the

supernatant, the pellet was resuspended in extraction buffer followed by the repetition of percoll steps. The re-suspended pellet from this round was saved as nuclear protein. Proteins were separated on 7.5% SDS-PAGE gel, transferred to PVDF membranes. Blots were probed with anti-phyB monoclonal B1 (1:300) or B1-7 (1:500) primary antibodies obtained from P. Quail lab. The secondary was anti-mouse horseradish peroxidase (HRP) (KPL, catalog no. 074-1806) (1:10,000). Chemiluminescent reactions were performed using the Pierce ECL Western Blotting Substrate following the manufacturer's instructions. The excessive substrate was removed from membranes before placing them between two plastic sheets to develop with X-ray, and subsequently scanned with Epson Perfection V600 Photo Scanner.

**SA extraction and analyses**. To extract SA, 50 mg of flash frozen plant tissue was ground in liquid nitrogen and extracted at 4 °C with 10% mM methanol and 1% acetic acid containing deuterated SA standard. Extracts were centrifuge twice and 20 μl of the clear supernatant was analyzed using a Dionex Ultimate 3000 binary RSLC system coupled to Thermo Q-Exactive Focus mass spectrometer with a heated electro spray ionization source. Sample separation was performed using a AcclaimTM RSLC 120 C18 column (100 × 2.1 mm, particle size 2.2 μM; Thermo Scientific 068982). Gradient elution was acetonitrile containing 0.1% formic acid followed with water containing 0.1% formic acid at a flow rate of 200 μL/min. The column temperature was maintained at 35 °C. Mass spectra were acquired in negative mode. Retention time and mass transitions was monitored on purified standard and plant matrixes. For relative quantitation, peak area for each compound (MS2; Thermo Trace Finder Software) was normalized to the initial fresh weight mass.

**MEcPP extraction and analyses**. To extract MEcPP, ~50 mg of flash frozen tissue was ground in liquid nitrogen and extracted at 4 °C with 10% mM methanol with 1% acetic acid, containing 10 μM of the deuterated standards. Extract was centrifuged twice and 100 μl of the clear supernatant was freeze dried and subsequently reconstituted in 500 μl acetonitrile–10 mM ammonium acetate (80:20, v/v). Ten microliters of the resuspended solution was injected to LC-MS system, and analyzed using a Dionex Ultimate 3000 binary RSLC system coupled to Thermo Q-Exactive Focus mass spectrometer with a heated electro spray ionization source. Sample was separated on an Accucore-150-Amide-HILIC column (150 × 2.1 mm; particle size 2.6 μM; Thermo Scientific 16726-152130) with a guard column containing the same column matrix (Thermo Scientific 852-00; 16726-012105). Gradient elution was performed with acetonitrile (A) and 10 mM ammonium acetate pH 7.0 (B). The separation was conducted using the following gradient profile (*t* (min), %A, %B): (−2, 90, 10), (0, 90, 100), (12, 30, 70), (16, 90, 10), (22, 90, 10). Flow rate was maintained at 280 μL/min. The column was kept at 35 °C. Mass spectra were acquired in negative mode and compound of interest was identified by accurate mass measurements (MS1), retention time and mass transitions of purified standard and plant matrixes.

**Sequencing and bioinformatics analysis**. RNA-seq libraries were constructed following BrAD-seq method[47] with six biological replicates. Seventy-five bases of single-end reads of the libraries were sequenced using the Illumina HiSeq 4000 at UC Davis DNA Technologies Core (United States). TopHat2 was utilized to map the reads to A. *thaliana* genome after trimming adaptors[48]. Subsequently DESeq2 was carried out to count and normalize the mapped reads[49]. Genes with >2-fold expression level change relative to control and *P*-value ≤ 0.05 were selected as differentially expressed genes (DEGs), which was further analyzed using AgriGO (http://bioinfo.cau.edu.cn/agriGO/analysis.php) to identify enriched Gene Ontology (GO) terms base on biological process category.

**Statistical analyses**. All experiments were performed with at least three biological replicates. Data are mean ± standard deviation (SD). The analyses were carried out via a two-tailed Student's *t* tests with a significance of $P < 0.05$.

**Reporting summary**. Further information on research design is available in the Nature Research Reporting Summary linked to this article.

## Data availability
RNA-seq data generated as part of this study has been deposited at the NCBI Sequence Read Archive with accession code PRJNA545720. The overlap between CAMTA3-, IAA-, PIF-, and *YHB*-regulated gene lists is provided in Supplementary Data 1–5. The source data underlying Figs. 1(c, d), 2(a–c), 3(a, b), 4(a, c–e), 5(b), 6(b), 7(b, d) and 8(a, b) are provided as Source Data file. The source data underlying Supplementary Figs. 1(a–c), 2(a, b), 4(a–c) and 5(a–c) are provided as Source Data file. Any additional data or biological material that support the findings of this study are available from the corresponding author upon reasonable request.

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

## Acknowledgements

We would like to thank Dr. Peter Quail (UC Berkeley) for providing phyB antibody and transgenic PIF lines, Dr. Akira Nagatani (Kyoto University) for providing phyB complementation construct, Dr. Meng Chen (UC Riverside) for allowing us to use his light chambers, and Mr. Derrick R. Hicks (University of Washington) for providing IAA and MEcPP structures for our model. We also would like to thank Dr. Jin-Zheng Wang for his comments on the paper. This work was supported by National Science Foundation (NSF) IOS-1036491, NSF IOS-1352478, and National Institutes of Health (NIH) R01GM107311 to K.D.

## Author contributions

J.J. and K.D. designed the study, J.J., L.Z., H.K. and B.D.L.C. performed the experiments, and K.D. wrote the paper.

## Additional information

**Competing interests:** Authors declare no competing interest.

