## [Peer Review File · Nature Communications]

Reviewers' comments:

Reviewer #1 (Remarks to the Author):

This manuscript "Orthogonal regulation of photomorphogenesis by stress-specific plastidial retrograde signaling metabolite" by Jiang et al. presents findings that reveal the signal transduction pathway of the stress retrograde signal MEcPP. The authors report the results of a genetic screen that identified two independent alleles of phyB as suppressors of the *ceh1* mutation which leads to increased accumulation of MEcPP and SA as well as severe stunting. They go on to demonstrate that phyB protein levels are elevated in plants with high MEcPP levels and that *ceh1* plants have a concomitant increase in sensitivity to red light. Most convincing for the MEcPP-phyB relationship, the authors showed that overexpression of PIF4 or PIF5 in the *ceh1* mutant background led to reduced phyB accumulation and partial recovery of the stunting and elevated SA level phenotypes of *ceh1* mutants. Consistent with previous reports, the authors also show that the phytohormone auxin and the CAMTA3 transcriptional regulator are important for regulating phyB protein levels.

The experiments are well done and the results are carefully presented. The Methods section largely provides enough detail for the repetition of the experiments described (see below for exception).

In total, these novel results considerably advance our understanding of the signaling circuits involved in control plant development in response to a multitude of environmental signals. They show that retrograde signaling via the MEcPP metabolite helps to determine phyB-mediated responses to light signals. These results should serve as a foundation for numerous follow-up studies about the relationship between retrograde signaling and photoreceptors. Further, these results further cement the idea of the chloroplasts as critical centers for the integration of sensing and responding to the environment.

Comments:

1. The suppressor screen is not described anywhere in the manuscript. These details should be included in the methods section of the manuscript.
2. A scale bar is provided in Fig. 7a, and similarly Fig. 1a and 4b need scale bars in order for claims about stunting reversal to be convincing.
3. Instead of referring to "robustly" repressed or down-regulated genes, the actual fold-change in expression should be stated. This is relevant for page 8 of the main text and Fig. 6.
4. The text of the manuscript has to be proofread and edited for small grammatical errors that make it harder to read the manuscript. One example from page 5 is "These data collectively verify revertants as phyB mutant alleles and establish selected function of phyB in MEcPP-mediated signal transduction pathways." What does "selected function of phyB" mean?

Tessa Burch-Smith

Reviewer #2 (Remarks to the Author):

Jiang and colleagues investigated the molecular connections of the stress-specific plastidial signal MEcPP with photomorphogenesis. A suppressor screen of *ceh1*, a high MEcPP containing mutant, identified two phyB mutant alleles, and they convincingly show that high MEcPP leads to elevated levels of phyB. They performed transcriptional, genetic and protein analyses to investigate how

MEcPP might be affecting phyB levels. Collectively, they propose a model in which the previously described alteration of auxin and CAMTA3 levels in *ceh*, and the newly found reduced PIF transcript levels, all regulate phyB levels via independent pathways. This is a timely and exciting finding that deepens our understanding of how stress-induced retrograde signaling impacts growth. However, in my opinion, the work would benefit from more direct evidence to support how phyB levels might be altered. Below are some comments and suggestions that might help improve it:

- The role of CAMTA3 and auxin in the regulation of phyB abundance is not investigated. The possibility that CAMTA3 acts at the proteasome level requires further exploration. Do CAMTA3 and auxin act independently from each other? Are they affecting PIF abundance or function? Although addressing these questions in detail might be the subject of future work, adding some more mechanistic insight would support the model and significantly add to the manuscript.
- The reduction in PIF transcript levels is quite small and might not be relevant at the protein level. I wonder if this really has an impact on phyB levels in *ceh1*.
- The effect of externally applied auxin suggests that altered levels of auxin in *ceh1* impact phyB levels. Genetic evidence would be necessary to provide support for this finding.
- The transcriptomic analysis of *camta3* suggests that there might be overlap with the phyB and PIF regulated gene network, and with hormone regulated pathways. It would be interesting to include comparisons with PIF and auxin regulated transcriptome that might suggest connections that are not explored now.

Other points:

- The reduction in SA levels in *cehphyB* is striking and would be interesting to discuss.
- Fig 2a, 2b, 4a, 4e: The light conditions and sampling are not detailed. Are they all in LD? What time were samples collected?
- Please describe "orthogonal regulation" in the context of this work.
- References need to be corrected. Citation of at least a few is incorrect and some relevant ones seem to be missing.

Reviewer #3 (Remarks to the Author):

It is known, that chloroplast retrograde and light signaling use common components, and that the chloroplast impacts on light-induced seedling development. In this paper, Jiang et al. try to disentangle the interconnections of the chloroplast retrograde signaling metabolite MEcPP and photomorphogenesis. In the current study under review, a suppressor screen of *ceh1* recovered two phyB mutant alleles which revert the *ceh1* dwarf phenotype. Subsequent experiments show that the *ceh1* mutant contains higher phyB protein levels compared to the WT. Thus it was hypothesized that the dwarf phenotype might be suppressed by lowering phyB protein levels in the *ceh1* mutant. Indeed, the *ceh1* dwarf phenotype is partially reverted, when the phyB levels are suppressed by (i) introducing *oePIF4* or *oePIF5* into *ceh1*, (ii) treating *ceh1* with auxin, (iii) crossing *ceh1* with the *camta3* mutant. Moreover, very moderate phyB protein repression was achieved by (iv) treating *ceh1* with an Ca²⁺ chelator or a Ca²⁺ channel blocker.

More specific comments:

Fig. 4: Overexpression of PIF4 and PIF5. Small differences of transcript accumulation measured in a real-time PCR experiment are not strongly supporting a downregulation. Moreover, the phyB-PIF interaction feedback-loop is leading to PIF PROTEIN degradation, PIF protein levels should be definitely measured here.

Panel e. It is known that overexpression of PIF4 or PIF5 reduces phyB protein level. Thus emphasis should be on the extent of phyB protein level reduction in the *ceh1* mutant. Numbers on phyB levels should be implemented in the figure.

Fig. 5: Which auxin concentration was used?

Fig. 6. RNA-Seq analysis.

„Detailed analyses of the data determined that the expression of ~90% of robustly repressed genes in the *ceh1* are CAMTA3-dependent, and are significantly overrepresented by genes in light- and auxin-response pathways (Fig. 6ab).“

This is not a „detailed“ analysis. The identified GOs are interesting. Which photosynthesis genes

are down-regulated? Why are those GOs, which include responses to light representative for the DOWN-regulated genes since phyB protein level are up in the *ceh1* mutant? How is the intersection with published phyB gain-of-function transcript data?

The regulation of the RSRE cis element is CAMTA3-dependent and was identified in induced genes in the *ceh1* mutant (Benn et al., PNAS 2016). Could you please comment also on the UP-regulated genes for the current RNA-Seq data?

Scale for the GO analysis: Log10 p-value: are the values in minus? Otherwise the p-values would be very high.

Fig. 7:

„In addition, we have shown partial size recovery of *ceh1* dwarf stature in the *ceh1/camta3* double mutant lines.“

The authors cite their literature. However, it is sometimes not see whether results shown in this manuscript were already produced in former publications. Clearly, Fig. 7a was already produced in Benn et al. and it should be more obvious.

“The data illustrate compromised growth phenotypes in seedlings treated with these two agents most notably in P plants.“

This result is not meaningful for the establishment of a connection between phyB protein abundance and growth behaviour of *ceh1*. The *ceh1* mutant does not look better under these treatments than under normal growth conditions.

„The result collectively supports regulatory function of CAMTA3 in stabilizing phyB protein levels in a Ca²⁺-dependent manner.“

This is not very clearly demonstrated because the reduction of phyB in the calcium experiment is only by 20%.

Fig. 8. Model. phyB protein level regulation by CAMTA3 and calcium is revealed by the *ceh1* mutant system. Thus, this model should definitely indicate that a block in HDS enzyme activity is leading to elevated MEcPP levels which in turn trigger subsequent

Overall:

- The title should be re-written, because it is misleading
- The language of the abstract and some other passages is of poor quality.
- Statistical significance testing should be indicated in the figures.
- One common theme of the *ceh1/phyB*, *ceh1/oePIF* and *ceh1/camta3* mutants is the reduction of salicylic acid levels compared to *ceh1*. The performance of a *ceh1* mutant in which the salicylic-acid pathway was interrupted should be investigated. With the *ceh1/eds16* mutant (Bjornson et al., Plant J 2017) the authors already have one suitable double mutant at hand.
- Further investigation of *ceh1/phyB*, *ceh1/oe-PIF*, *ceh1/camta3* seedlings and the addition of Ca²⁺ to *ceh1* is needed:

a) The introduction section has one focus on SAR and the title is about photomorphogenesis. To underpin the statement “whereby a specific plastidial signal such as MEcPP is transduced into discrete biological responses that collectively regulate photomorphogenesis”, it should be investigated at least whether in addition to the dwarf phenotype of the *ceh1* mutant the hypocotyl-length phenotype of *ceh1* is reverted? How is flowering time altered?

b) In the model and the discussion section it is proposed that the “function of CAMTA3 is likely via impairment of proteasome-mediated degradation of the light-activated photoreceptor proteasome COP1-mediated proteolysis of phyB, achieved either through direct targeting of a component(s) of COP1 machinery and/or indirectly via targeting carbon status.” This would be an important new

key finding which should be definitely further investigated in this manuscript. In *ceh1*, phyB protein levels in the light stay as high as in darkness which is reminiscent of the *cop1* mutant, indicating that proteasome-mediated degradation of phyB in *ceh1* might indeed not be functional. The phyB Western blots of the *ceh1* and *ceh1/camta3* mutants and the auxin treated plants should also be shown in the presence of the proteasome inhibitor MG132.

The postulated phyB degradation by CAMTA3 should also be investigated by an in vitro assay similar to that shown in Figure 6, Jang et al., Plant Cell 2010

Minor comments:

The Nam-Hai Chua group should be cited with their Jang et al, Plant Cell 2010, manuscript in which phyB polyubiquitination by COP1 in the nucleus is described.

Dear reviewers to address your collective concerns we have performed almost all of the additional experiments that are pertinent and within the scope of the current work. We truly believe that compliance with your collective inputs and suggestions has improved the overall quality of the manuscript and has brought to attention the central theme not previously discussed. It should be noted that because of the substantial alterations, the changes are not highlighted in the text.

The specific response to each of your concerns are as follows:

Reviewer #1

1. The suppressor screen is not described anywhere in the manuscript. These details should be included in the methods section of the manuscript.

This information is now included in the method section.

2. A scale bar is provided in Fig. 7a, and similarly Fig. 1a and 4b need scale bars in order for claims about stunting reversal to be convincing.

Scale bar is now provided in all the plant images (Figs. 1 a-b, 4b, 5a, 6a, 7a, and c).

3. Instead of referring to “robustly” repressed or down-regulated genes, the actual fold-change in expression should be stated. This is relevant for page 8 of the main text and Fig. 6.

We have resolved this issue throughout the manuscript.

4. The text of the manuscript has to be proofread and edited for small grammatical errors that make it harder to read the manuscript. One example from page 5 is “These data collectively verify revertants as phyB mutant alleles and establish selected function of phyB in MEcPP-mediated signal transduction pathways.” What does “selected function of phyB” mean?

We have edited the entire manuscript and rewritten some sections including the one this reviewer is referring to. The text now reads: “The data collectively establishes phyB as a component of MEcPP signal transduction pathway involved in regulation of growth and SA level, but not in the induction of *HPL* expression.

Reviewer #2

1. The role of CAMTA3 and auxin in the regulation of phyB abundance is not investigated. The possibility that CAMTA3 acts at the proteasome level requires further exploration.

We have established the likely function of CAMTA3 in impairing the degradation machinery by examining phyB protein levels in mock and bortezomib treated seedlings (Figs. 8a-c & S5a-c).

2. Do CAMTA3 and auxin act independently from each other?

We had previously established that CAMTA3 does not alter auxin levels (Benn et al, PNAS 2016). Here we show the opposing functions of the two; CAMTA3 is stabilizing PhyB protein (Fig. 7a) whereas auxin reduces the levels (Figs. 5b and 6b).

3. Are they affecting PIF abundance or function?

We have not yet explored this possibility, but we will in future.

4. The reduction in PIF transcript levels is quite small and might not be relevant at the protein level. I wonder if this really has an impact on phyB levels in ceh1.

We agree with the reviewer's comment, however our genetic approach using PIF over-expressers has confirmed the role of PIFs in regulating phyB protein abundance in the *ceh1* mutant background.

5. The effect of externally applied auxin suggests that altered levels of auxin in ceh1 impact phyB levels. Genetic evidence would be necessary to provide support for this finding.

To address this concern we have generated *ceh1/tir1* double mutant line and grown them along with their respective parent backgrounds in the presence and absence of auxin. The results clearly show negative regulatory function of auxin in regulating phyB abundance (Fig. 6a-b)

6. The transcriptomic analysis of camta3 suggests that there might be overlap with the phyB and PIF regulated gene network, and with hormone regulated pathways. It would be interesting to include comparisons with PIF and auxin regulated transcriptome that might suggest connections that are not explored now.

We have performed bioinformatics analyses and presented the Venn diagrams along with the list of overlapping genes between *ceh1*, CAMTA3, PIF, IAA treated, and constitutive active allele of PhyB (*YHB*) (Figs. 3a-e and Table. S1-5).

Other points:

-The reduction in SA levels in ceh/phyB is striking and would be interesting to discuss.

We have briefly discussed it in discussion section.

-Fig 2a, 2b, 4a, 4e: The light conditions and sampling are not detailed. Are they all in LD? What time were samples collected?

The pertinent information is now included.

-Please describe "orthogonal regulation" in the context of this work.

MEcPP-mediates multifaceted and intersecting regulatory networks that regulate the abundance of the red light photoreceptor, phyB, via auxin signalling cascade and Ca²⁺-dependent CAMTA3-based transcriptional network.

-References need to be corrected. Citation of at least a few is incorrect and some relevant ones seem to be missing.

We apologize for this embarrassing oversight. We were not aware of the glitch in our Endnote system. The references are now corrected. Thanks for pointing out this important error.

Reviewer #3

*1. Fig. 4: Overexpression of PIF4 and PIF5. Small differences of transcript accumulation measured in a real-time PCR experiment are not strongly supporting a downregulation. Moreover, the phyB-PIF interaction feedback-loop is leading to PIF PROTEIN degradation, PIF protein levels should be definitely measured here. Panel e. It is known that overexpression of PIF4 or PIF5 reduces phyB protein level. Thus emphasis should be on the extent of phyB protein level reduction in the *ceh1* mutant. Numbers on phyB levels should be implemented in the figure.*

Our several attempts to examine PIF protein levels in *ceh1* has proven unsuccessful. We agree with the reviewer's comments also shared by the reviewer #2. As such, we resorted to a genetic approach using *ceh1/PIF* overexpressers to overcome this shortcoming. We have included number on phyB immunoblots (Figs. 4e, 5b, 6b, 8a-b and S5a-b).

2. Fig. 5: Which auxin concentration was used?

We apologize for this omission, the concentrations are now included in the corresponding figure legends.

3. Fig. 6. RNA-Seq analysis.

*Detailed analyses of the data determined that the expression of ~90% of robustly repressed genes in the *ceh1* are CAMTA3-dependent, and are significantly overrepresented by genes in light- and auxin-response pathways (Fig. 6ab).*

*This is not a "detailed" analysis. The identified GOs are interesting. Which photosynthesis genes are down-regulated? Why are those GOs, which include responses to light representative for the DOWN-regulated genes since phyB protein level are up in the *ceh1* mutant? How is the intersection with published phyB gain-of-function transcript data?*

*The regulation of the RSRE cis element is CAMTA3-dependent and was identified in induced genes in the *ceh1* mutant (Benn et al., PNAS 2016). Could you please comment also on the UP-regulated genes for the current RNA-Seq data?*

Scale for the GO analysis: Log10 p-value: are the values in minus? Otherwise the p-values would be very high.

We have fully complied with the reviewer's suggestion, altered the text and included additional bioinformatics in supplemental data (Fig.s3a-e and Table S1-5).

4. Fig. 7: *“In addition, we have shown partial size recovery of ceh1 dwarf stature in the ceh1/camta3 double mutant lines.”*

The authors cite their literature. However, it is sometimes not see whether results shown in this manuscript were already produced in former publications. Clearly, Fig. 7a was already produced in Benn et al. and it should be more obvious.

The images shown here **were not** from the PNAS paper 2016, but the reconfirmation of the data by repeating the experiments. We have now made it fully clear in the text.

5. *“The data illustrate compromised growth phenotypes in seedlings treated with these two agents most notably in P plants.”*

This result is not meaningful for the establishment of a connection between phyB protein abundance and growth behaviour of ceh1. The ceh1 mutant does not look better under these treatments than under normal growth conditions.

Fig 7c shows relative growth of P versus *ceh1* plant in the presence and absence of LaCl₃. In our view the image clearly shows that the agent negatively impacts the growth of P plants more notably than that of the *ceh1*. Hence, we believe that this image supports our claim.

6. *“The result collectively supports regulatory function of CAMTA3 in stabilizing phyB protein levels in a Ca²⁺-dependent manner.”*

This is not very clearly demonstrated because the reduction of phyB in the calcium experiment is only by 20%.

We have repeated the experiment with a higher LaCl₃ concentration. The current result shows clear reduction in phyB protein levels (Fig. 7d).

7. Fig. 8. *Model. phyB protein level regulation by CAMTA3 and calcium is revealed by the ceh1 mutant system. Thus, this model should definitely indicate that a block in HDS enzyme activity is leading to elevated MEcPP levels which in turn trigger subsequent.*

We have complied with the reviewer's suggestion, see Fig. 9.

Overall:

- The title should be re-written, because it is misleading

As suggested we have modified the title and replaced photomorphogenesis with phyB

The language of the abstract and some other passages is of poor quality.

The abstract and many passages are rewritten.

- *Statistical significance testing should be indicated in the figures.*

We have complied, please see fig legends.

- *One common theme of the *ceh1/phyB*, *ceh1/oePIF* and *ceh1/camta3* mutants is the reduction of salicylic acid levels compared to *ceh1*. The performance of a *ceh1* mutant in which the salicylic-acid pathway was interrupted should be investigated. With the *ceh1/eds16* mutant (Bjornson et al., *Plant J* 2017) the authors already have one suitable double mutant at hand.*

This data was already included, please see Fig. S2a.

-*Further investigation of *ceh1/phyB*, *ceh1/oe-PIF*, *ceh1/camta3* seedlings and the addition of Ca^{2+} to *ceh1* is needed:*

*a) The introduction section has one focus on SAR and the title is about photomorphogenesis. To underpin the statement “whereby a specific plastidial signal such as MEcPP is transduced into discrete biological responses that collectively regulate photomorphogenesis”, it should be investigated at least whether in addition to the dwarf phenotype of the *ceh1* mutant the hypocotyl-length phenotype of *ceh1* is reverted? How is flowering time altered?*

We have now included images and measurements of the hypocotyl length (see Fig. 1b and S1a). In regards to the flowering time we have already published the early flowering phenotype of *ceh1* in: Wang, et al, *The Plant Cell* (2014); and Wang, C, and Dehesh, K., *Plant Signaling & Behavior* 10(6): e1022012 (2015). From retrograde signaling to flowering time.

Lastly, we disagree with the reviewer’s comment concerning adding Ca^{2+} to *ceh1*. We don’t see any value in performing this experiment, nor the purpose and the conclusion one can draw from it.

*The *phyB* Western blots of the *ceh1* and *ceh1/camta3* mutants and the auxin treated plants should also be shown in the presence of the proteasome inhibitor MG132.*

We have complied and established the likely function of CAMTA3 in impairing the degradation machinery by examining phyB protein levels in mock and bortezomib treated seedlings (Figs. 8a-c & S5a-c).

*The postulated *phyB* degradation by CAMTA3 should also be investigated by an *in vitro* assay similar to that shown in Figure 6, Jang et al., *Plant Cell* 2010.*

We disagree with this suggestion, since an *in vitro* assay is outside the current scope of this work and not necessary.

Minor comments:

The Nam-Hai Chua group should be cited with their Jang et al, Plant Cell 2010, manuscript in which phyB polyubiquitination by COP1 in the nucleus is described.

We agree and now we have included this important reference, sorry for the oversight.

REVIEWERS' COMMENTS:

Reviewer #1 (Remarks to the Author):

The revised version of the manuscript by Jiang and co-workers presents strong evidence that the plastidial retrograde signal MEcPP can act on phytochrome B levels to mediate plant responses to the environment. The interplay between chloroplast signaling and light perception pathways in the nucleus and cytoplasm is of fundamental importance to understanding how plants sense and respond to their environment. This paper enhances our understanding of this crosstalk and how these signals can be integrated to produce a coherent outcome. The revisions made by the authors have improved the clarity of work. In particular, the addition of details of the genetic screen and a more thorough discussion of the gene expression analysis along with the inclusion of experiments demonstrating the importance of proteasome-mediated degradation in the regulation of PhyB have made the story more complete. I only have a few comments:

1. Figure 1 a and b panels don't line up and since the labels provided better align to a than b, then the figures are hard to follow. Further, there is an extra dividing line in a in the phyB-9 images, adding to the confusion.
2. The relationship between Figure 8 and Figure S5 was not clear to me. Were they showing 4 replicates of the same experiment to illustrate the variability of the assay but yet consistent recovery of phyB in the presence of the proteasome inhibitor?
3. The final paragraph in the Discussion argues that the work demonstrates the relationship between MEcPP and photomorphogenesis, yet only mature plants are used for the present analysis. It is more correct to say that they are examining light-perception and chloroplast retrograde signaling, as was done in the Introduction. "Photomorphogenesis" would be more appropriate if younger stages of plant development like seedlings.

Reviewer #2 (Remarks to the Author):

Authors have addressed all my concerns satisfactorily, I appreciate their efforts.

Reviewer #3 (Remarks to the Author):

The authors addressed most of my concerns. However, this reviewer is concerned about the equal phyB protein level in mock-treated *ceh1* and *ceh1 camta3* seedlings (Fig. 8b) which would argue against the assumption of CAMTA3-mediated stabilization of the phyB protein.

Response to the Reviewers:

We would like to express our sincere thanks to all the three reviewers for their efforts. Our response to their latest comments is as follows:

Reviewer#1:

1. Figure 1 a and b panels don't line up and since the labels provided better align to a than b, then the figures are hard to follow. Further, there is an extra dividing line in a in the phyB-9 images, adding to the confusion.

We have altered the presentation of the images to the possible extent, however size differences amongst seedlings interfere with the full alignment. However, we believe that labeling at the top and bottom of the panels would overcome the confusion. In addition, the dividing line in the *phyB-9* image is removed.

2. The relationship between Figure 8 and Figure S5 was not clear to me. Were they showing 4 replicates of the same experiment to illustrate the variability of the assay but yet consistent recovery of phyB in the presence of the proteasome inhibitor?

We have included the two representative blots, both of which support the consistent response of seedlings to the proteasome inhibitor (bortezomib), albeit at different degrees potentially due to differential penetration of bortezomib. Please also see our response to the reviewer#3.

3. The final paragraph in the Discussion argues that the work demonstrates the relationship between MEcPP and photomorphogenesis, yet only mature plants are used for the present analysis. It is more correct to say that they are examining light-perception and chloroplast retrograde signaling, as was done in the Introduction. "Photomorphogenesis" would be more appropriate if younger stages of plant development like seedlings.

We respectfully disagree with the reviewer's comment, since we have included images of young seedlings to make the point that the growth phenotypes and by extension molecular phenotypes encompass all developmental stages, including younger stages of development (See Fig. 1a). Accordingly we justify the usage of the language "photomorphogenesis".

Reviewer #2:

Authors have addressed all my concerns satisfactorily, I appreciate their efforts.

Thanks.

Reviewer #3:

The authors addressed most of my concerns. However, this reviewer is concerned about the equal phyB protein level in mock-treated ceh1 and ceh1 camta3 seedlings (Fig. 8b) which would argue against the assumption of CAMTA3-mediated stabilization of the phyB protein.

We would like to emphasize the fact that the mock treatment here reflect seedling's treatment with DMSO, the solvent for bortezomib. We believe that there is variation in the degree of bortezomib penetration, and in addition the presence of DMSO in mock treated seedlings further enhances the variation in response. For this specific purpose we have repeated the experiments several times and here we included the two representative blots namely the one in Fig 8 (Supplemental Fig 5 in the previous version) and the other in Fig Supplementary Fig. 5. All blots including the ones presented support consist response of seedlings to the proteasome inhibitor (bortezomib), albeit at different degrees potentially due to differential penetration of bortezomib. In response to the reviewer's comment we have inserted the following language in the manuscript shown in highlighted text "It should be noted that both presented blots consistently display higher phyB protein levels in the presence of proteasome inhibitor albeit at different degrees, most likely due to varying bortezomib penetration rate."